# Association between meat intake and mortality due to all-cause and major causes of death in a Japanese population

Eiko Saito[1,2], Xiaohe Tang[3], Sarah Krull Abe[2], Norie Sawada[2], Junko Ishihara[4], Ribeka Takachi[5], Hiroyasu Iso[6], Taichi Shimazu[2], Taiki Yamaji[2], Motoki Iwasaki[2], Manami Inoue[2]*, Shoichiro Tsugane[2], for the JPHC Study Group[¶]

1 Division of Cancer Statistics Integration, Center for Cancer Control and Information Services, National Cancer Center, Chuo-ku, Tokyo, Japan, 2 Epidemiology and Prevention Group, Center for Public Health Sciences, National Cancer Center, Chuo-ku, Tokyo, Japan, 3 Department of Global Health Policy, Graduate School of Medicine, The University of Tokyo, Bunkyo-ku, Tokyo, Japan, 4 Department of Food and Life Science, School of Life and Environmental Science, Azabu University, Sagamihara-shi, Kanagawa, Japan, 5 Department of Food Science and Nutrition, Faculty of Human Life and Environment, Nara Women's University, Nara-shi, Nara, Japan, 6 Public Health, Department of Social and Environmental Medicine, Osaka University Graduate School of Medicine, Suita-shi, Osaka-fu, Japan

¶ Membership of the JPHC Study Group is provided in the Acknowledgments.
* mnminoue@ncc.go.jp

**Data Availability Statement:** Regarding data accessibility, we cannot publicly share the individual data even after anonymization according to the ethical guidelines in Japan if: a) consent to

## Abstract

### Purpose

We examined the association between meat intake and mortality due to all-cause and major causes of death using a population-based cohort study in Japan.

### Methods

87,507 Japanese aged between 45 and 74 years old at 5-year follow-up study were followed for 14.0 years on average. Associations between meat intake and mortality risk were assessed using a Cox proportional hazards model.

### Results

A heavy intake of total meat was associated with a higher risk of all-cause mortality relative to the lowest quartile intake in men (Q4: HR,1.18; 95%CIs, 1.06–1.31). A higher intake of total meat was associated with a lower risk of stroke mortality in women (Q2: HR, 0.70; 95% CIs, 0.51–0.94, Q3: HR, 0.68; 95%CIs, 0.50–0.95, Q4: HR, 0.66; 95%CIs, 0.44–0.99). A heavy intake of red meat was also associated with all-cause mortality (Q4: HR, 1.13; 95% CIs, 1.02–1.26) and heart disease mortality (Q4: HR, 1.51; 95%CIs, 1.11–2.06) in men but not in women. Heavy intake of chicken was inversely associated with cancer mortality in men.

public sharing of data has not been obtained from each participant, and b) there is a possibility of identifying an individual by combining the information or due to the existence of rare diseases, etc. Therefore, the investigators require the approval of the JPHC Steering Committee (SC) and the Institutional Review Board (IRB). Currently, only citizens of Japan who fulfill the requirements of conducting research projects are eligible to apply for the JPHC data and/or biospecimens. The investigator is required to submit a Project Protocol (Research question, Aim, Background, Design and analytical plan) for review by the JPHC SC. Requests can be made by contacting the JPHC SC directly (jphcadmin@ml.res.ncc.go.jp) or investigators in the JPHC. After approval is obtained from the JPHC SC, the investigator is required to obtain additional approval for the protocol under which the data and/or biospecimens are to be used from the Institutional Review Board of the National Cancer Center. This information on how to submit an application to gain access to JPHC data/or biospecimens is publicly available at http://epi.ncc.go.jp/en/jphc/805/8155.html.

**Funding:** This study was supported by National Cancer Center Research and Development Fund (23-A-31[toku], 26-A-2 and 29-A-4) (since 2011) and a Grant-in-Aid for Cancer Research from the Ministry of Health, Labour and Welfare of Japan (from 1989 to 2010). The funders had no role in study design, data collection and analysis, decision to publish, or preparation of the manuscript.

**Competing interests:** The authors have declared that no competing interests exist.

## Conclusions

Heavy intakes of total and red meat were associated with an increase in all-cause and heart disease mortality in men, while total meat intake was associated with a lower risk of stroke mortality in women.

## Introduction

Meat is a major source of protein and fat in the diet of many countries around the world [1]. Total meat intake is generally reported to be considerably higher in Western countries than in Asia. For instance, a study using the Food and Agriculture Organization (FAO) database showed that per capita consumption of total meat in the United States was approximately three times higher than that in Japan, China and South Korea since the 1990s [2]. Indeed, mean consumption of meat in the United States in 2007 was 122.8 kg per year versus 46.1 to 55.9 kg in these three countries [2].

In Japan, westernization of the diet has seen a near-doubling of meat consumption between 1970 and 2006 [2]. Increased intake of animal fat and/or protein has contributed to reducing stroke among Japanese since the 1960s [3]. However, detrimental effects of meat consumption have also been reported, albeit primarily from Western populations. Epidemiological studies have reported that excessive intake of meat, especially red meat and processed meat, is associated with increased risk of morbidity, including heart disease [4], cardiovascular disease [5], diabetes [6, 7] and certain types of cancer [8, 9]. Elucidating the association of meat intake with overall mortality will aid in assessing the differential impact of meat intake on health among Asians. Current reporting of summary estimates of the link between meat consumption and mortality has primarily been obtained from Western populations [10–13], where meat intake is considerably higher than that in Asian populations. Evidence from Japan is scarce: only a few studies have reported the association of meat intake on mortality [2, 3, 14, 15] and the results are in any case disconcordant. A recent study in 2019 published from the same Japanese cohort reported associations of protein intake with mortality [16]; however, the study primarily assessed the effect of animal and plant-based protein intake on mortality, and did not provide a comprehensive breakdown of item-specific associations.

Here, we aimed to investigate the association between meat intake and all-cause and major causes of death, including cancer, heart disease and cerebrovascular disease in Japan using comprehensive, item-specific information obtained from a large-scale, prospective cohort study.

## Methods

### Study population

The baseline study for Cohort I started in 1990 and that for Cohort II in 1993, covering a total of 140,420 participants (68,722 men and 71,698 women) in 11 public health center areas. The study enrolled participants aged 40 to 59 years in Cohort I and 40 to 69 years in Cohort II. Details of the Japan Public Health Center-based Prospective Study have been described elsewhere [17, 18]. The study was approved by the Institutional Review Boards of the National Cancer Center (approval number: 2001–021) and The University of Tokyo (approval number: 10508). Participants in the current study were subjects in the JPHC study who were enrolled at age 45–74 years and who responded to a self-administered 5-year follow-up questionnaire,

which included comprehensive information on dietary intake and lifestyle-related factors, between 1995 and 1999. This follow-up survey was used as the starting point in the present study. After exclusion of participants who had died, migrated outside of Japan, or were lost to follow-up before the start of the 5-year follow-up survey, the remaining 125,363 subjects were eligible for participation. Of these, 99,629 subjects returned the completed questionnaire (response rate = 79%). Participants were further excluded if they reported a past history of heart disease, cancer or stroke at the time of the baseline or 5-year follow-up survey (n = 6,588), and those with missing data on variables including intakes of meat and total energy or who reported extreme caloric intake (upper and lower 2.5%) (n = 5,534). The final analytic cohort included 87,507 participants (40,072 men and 47,435 women) (S1 Fig).

## Follow-up

Participants were followed from the date of the 5-year follow-up survey to the date of death or to the end date of follow-up (December 31, 2011) except for Katsushika area, for which the follow-up was terminated on December 31, 2009. Subjects who died or moved to other areas were followed through the residential registry. Cause of death was ascertained by death certificates with permission of the Ministry of Health, Labour and Welfare [18]. Causes of death were classified using International Classification of Diseases and Related Health Problems, Tenth Revision (ICD-10) [19], namely cancer (C00-C97), heart disease (I20-I52), cerebrovascular disease (I60-I69), colorectal cancer (C18-C20) and all-cause mortality. We also included mortality from ischemic heart disease (I20-I25) and intracerebral hemorrhage (I61) in our sub-analysis.

## Assessment of exposures

Dietary information was collected through a validated self-administered food frequency questionnaire (FFQ) which enquires about the average frequency and portion size of 138 foods and beverages consumed in the past year [20, 21]. Red meat items included three beef dishes (steak, grilled beef and stewed beef), six pork dishes (stir-fried pork, deep-fired pork, stewed pork in western style, stewed pork in Japanese style, pork in soup, and pork liver), four processed meat products (ham, sausage or Wiener sausage, bacon and luncheon meat) and chicken liver. Chicken items included two chicken dishes (grilled chicken and deep-fried chicken). For each food item, nine response categories were provided to report consumption frequency, ranging from rarely (<1 time/month) to more than 7 times a day. The standard portion sizes were enquired about for each food item in three choices: small (less than half of the standard portion), medium (standard portion), and large (more than 1.5 times the standard portion). Participants were categorized by quartiles of total meat, red meat, beef, pork, processed meat and chicken consumption for men and women separately. The validity of the FFQ for the assessment of meat intake has been previously reported as fair to moderate [20, 21]. Spearman's correlation coefficients comparing energy-adjusted meat intake derived from the FFQ with one derived from 28-day (or 14-day) dietary records were 0.50 and 0.45 for men and women in Cohort I [20], and 0.48 and 0.44 for men and women in Cohort II [21], respectively. As for the reproducibility of the FFQ, correlation coefficients comparing the two FFQ values administered 1 year apart were 0.52 for Cohort I (men and women) [22] and 0.52 for men and 0.41 for women in Cohort II [20]. Spearman's correlation coefficients comparing the energy-adjusted intake of specific meats for men derived from the FFQ and that derived from 28-day (or 14-day) dietary records for men were as follows: beef, 0.43; pork, 0.42; processed meat, 0.45; and chicken, 0.20 [23].

## Statistical analysis

Person-years for each participant were calculated from the date of the 5-year follow-up survey until the date of death or the date of censoring (December 31, 2011), whichever occurred first. Cox proportional hazards regression models were used to estimate the Hazard Ratios (HRs) and 95% confidence intervals (CIs) between quartiles of meat intake and risk of all-cause and cause-specific mortality, with the lowest quartile category being the reference, and by modeling the risk factors separately for men and women. The model was adjusted for age (years, continuous); public health center area; cigarette smoking (never, past, current); alcohol consumption (none; drinker: <150, 150–299, 300+ grams of ethanol per week); body mass index (BMI) (<25, 25–27, 28-<30, 30+,); metabolic equivalent task-hours [24] per day (in quartiles); and history of diabetes or hypertension (yes, no). Missing values for each of these covariates were grouped into one and included in the analysis. We also adjusted the model for consumption of the following food items (in grams/day): fruit, vegetables, fish, dairy products, egg, sodium in addition to total fat and total energy intake. All food intakes were energy-adjusted for men and women separately using the residual method. Tests for linear trend were performed by assigning scores for each intake category for each type of meat intake, starting from one for the lowest consumption status of meat to four for the highest as a continuous variable. We repeated the same analysis after excluding deaths that occurred within 5 years after the 5-year follow-up survey to avoid potential bias from subclinical illnesses. Additionally, sub-group analyses by age groups at 5-year follow-up (45–54 years, 55–64 years, 65 years and older) were performed to assess the generational differences in the associations by using the fully-adjusted model. We computed P-interaction values by using likelihood ratio tests to compare Cox proportional hazards models with and without cross-product terms for meat intake (in scores) and age (45–54 years, 55–64 years, and 65 years and older) in the subgroup analyses. Proportional hazards assumptions were tested using Shoenfeld residuals and found to be nonsignificant. Throughout the paper, all p-values are two-sided, and statistical significance was set at smaller than p<0.05 level. All analyses were conducted using Stata SE 14 (StatCorp, College Station, TX).

## Results

During 1,225,257 person-years of follow-up (average 14.0 years), there were 9,886 deaths due to all-cause mortality (6,266 for men, 3,620 for women), 4,174 deaths due to cancer (2,695 for men, 1,479 for women), 940 deaths due to cerebrovascular disease (569 for men, 371 for women), and 1,209 deaths due to heart disease (751 for men, 458 for women). Table 1 shows the characteristics of participants according to quartiles of total meat intake by sex. Those who consumed more meat tended to be younger, and consumed less fruits, vegetables and dairy products.

Tables 2 and 3 show Hazard Ratios (HRs) and 95% confidence intervals (95% CI) for the association between meat consumption and all-cause and cause-specific mortality separately for men and women, and S1 and S2 Tables show the Hazard Ratios and 95% CIs after excluding deaths that occurred within five years after the 5-year follow-up survey. After adjusting for potential confounders, total meat intake was associated with a higher risk of all-cause mortality in men [Q1: reference; Q4: HR, 1.18 (95% CIs, 1.06–1.31)]. Total meat intake was also associated with an elevated risk of heart disease mortality only in men [Q1: reference; Q4: HR, 1.46 (95% CIs, 1.08–1.99)]. In contrast, an intake of total meat was associated with a lower risk of stroke mortality in women [Q2: HR, 0.70 (95% CIs, 0.51–0.94), Q3: HR, 0.68 (95% CIs, 0.50–0.95), Q4: HR, 0.66 (95% CIs, 0.44–0.99)].

Our analyses by meat type showed that a higher intake of red meat was also associated with all-cause mortality [Q1: reference; Q4: HR, 1.13 (95% CIs, 1.02–1.26)] and heart disease

**Table 1. Basic characteristics of participants by quartile of energy-adjusted total meat intake in the study population.**

| Characteristic | Quartiles of total meat intake | | | | | | | | | |
| --- | --- | --- | --- | --- | --- | --- | --- | --- | --- | --- |
| | Men | | | | | Women | | | | |
| | Q1 | Q2 | Q3 | Q4 | P-value[1] | Q1 | Q2 | Q3 | Q4 | P-value[1] |
| Total participants (n = 87,507) | 10,018 | 10,018 | 10,018 | 10,018 | | 11,859 | 11,859 | 11,859 | 11,858 | |
| Age (years), mean ±SE[2] | 58.5±0.1 | 57.3±0.1 | 56.8±0.1 | 57.0±0.1 | <0.001 | 59.0±0.1 | 57.7±0.1 | 57.2±0.1 | 57.3±0.1 | <0.001 |
| Total meat intake (g/d),[3] mean ±SE | 17.5±0.1 | 38.4±0.1 | 58.7±0.1 | 106.4±0.4 | <0.001 | 16.8±0.1 | 38.1±0.05 | 58.1±0.1 | 104.2±0.3 | <0.001 |
| Red meat intake (g/d), mean ±SE | 14.3±0.1 | 31.9±0.1 | 49.8±0.1 | 92.9±0.4 | <0.001 | 13.6±0.1 | 31.5±0.1 | 48.7±0.1 | 90.3±0.3 | <0.001 |
| Beef intake (g/d), mean ±SE | 5.5±0.1 | 12.7±0.1 | 20.7±0.1 | 38.2±0.3 | <0.001 | 3.1±0.04 | 7.9±0.1 | 12.6±0.1 | 23.1±0.2 | <0.001 |
| Pork intake (g/d), mean ±SE | 7.3±0.1 | 16.2±0.1 | 25.4±0.1 | 50.4±0.3 | <0.001 | 8.0±0.1 | 18.2±0.1 | 28.4±0.1 | 56.7±0.3 | <0.001 |
| Processed meat intake (g/d), mean ±SE | 1.3±0.02 | 2.9±0.03 | 4.5±0.05 | 8.4±0.1 | <0.001 | 2.1±0.03 | 4.6±0.04 | 6.9±0.1 | 11.7±0.1 | <0.001 |
| Chicken intake (g/d), mean ±SE | 2.8±0.03 | 5.7±0.04 | 8.1±0.1 | 13.4±0.1 | <0.001 | 3.2±0.03 | 6.6±0.05 | 9.5±0.1 | 15.1±0.1 | <0.001 |
| Current smoker (%) | 43.9 | 45.9 | 46.0 | 42.4 | <0.001 | 5.3 | 4.9 | 5.2 | 6.1 | <0.001 |
| Alcohol intake per week (g), mean ±SE | 263.4±3.1 | 224.7±2.6 | 191.1±2.3 | 133.3±1.9 | <0.001 | 18.0±0.8 | 16.3±0.6 | 13.6±0.5 | 10.7±0.4 | <0.001 |
| Body mass index (kg/m^2), mean ±SE | 23.4±0.03 | 23.5±0.03 | 23.6±0.03 | 23.8±0.03 | <0.001 | 23.4±0.03 | 23.4±0.03 | 23.4±0.03 | 23.7±0.03 | <0.001 |
| Physical activity (MET-h/d), mean ±SE | 33.2±0.1 | 33.1±0.1 | 32.8±0.1 | 32.2±0.1 | 0.070 | 32.0±0.05 | 32.2±0.05 | 32.1±0.05 | 31.6±0.05 | 0.773 |
| History of hypertension (%) | 24.4 | 22.5 | 21.4 | 21.0 | <0.001 | 24.5 | 21.5 | 20.6 | 21.8 | <0.001 |
| History of diabetes (%) | 8.8 | 8.3 | 8.3 | 8.1 | <0.001 | 4.4 | 3.9 | 3.8 | 4.0 | <0.001 |
| Total energy (kcal /d), mean ±SE | 2268.2±6.6 | 2248.1±6.1 | 2169.2±6.1 | 2002.1±6.1 | <0.001 | 1930.1±5.5 | 1914.1±4.9 | 1868.2±4.9 | 1741.3±5.0 | <0.001 |
| Other dietary intake[3] | | | | | | | | | | |
| Vegetables (g/d), mean ±SE | 127.5±1.0 | 126.7±0.8 | 124.3±0.8 | 124.4±0.8 | 0.010 | 242.5±1.4 | 228.8±1.1 | 216.0±1.0 | 201.3±1.0 | <0.001 |
| Fruits (g/d), mean ±SE | 86.5±0.8 | 86.7±0.7 | 83.2±0.7 | 75.4±0.6 | <0.001 | 273.7±1.8 | 250.5±1.5 | 226.7±1.4 | 190.8±1.4 | <0.001 |
| Fish (g/d), mean ±SE | 66.9±0.5 | 68.8±0.4 | 72.2±0.4 | 72.7±0.4 | <0.001 | 82.7±0.5 | 83.4±0.4 | 85.0±0.4 | 82.7±0.4 | <0.001 |
| Egg (g/d), mean ±SE | 26.6±0.3 | 26.5±0.3 | 27.4±0.3 | 29.4±0.3 | <0.001 | 25.9±0.3 | 27.2±0.2 | 28.2±0.2 | 29.2±0.2 | <0.001 |
| Dairy products (g/d), mean ±SE | 76.0±1.0 | 67.9±0.8 | 64.3±0.7 | 60.4±0.7 | <0.001 | 226.7±2.1 | 209.1±1.7 | 186.3±1.5 | 166.2±1.6 | <0.001 |
| Sodium (mg/d), mean ±SE | 3850.5±14.1 | 3918.1±12.5 | 3990.6±12.4 | 4109.2±12.5 | <0.001 | 4458.3±14.1 | 4430.5±50.1 | 4399.8±11.8 | 4468.8±43.3 | <0.001 |
| Total fat (g/d), mean ±SE | 33.1±0.1 | 38.3±0.1 | 43.3±0.1 | 54.5±0.1 | <0.001 | 45.5±0.1 | 50.6±0.1 | 55.1±0.1 | 54.5±0.1 | <0.001 |

Abbreviations: Q, quartile; MET, metabolic equivalents.

[1] ANOVA or chi-square test.

[2] SE, Standard errors.

[3] All mean total intakes of food and nutrients are energy-adjusted.

mortality [Q1: reference; Q4: HR, 1.51 (95% CIs, 1.11–2.06)] in men but not in women. Among different types of red meat, beef consumption was associated with an increased risk of cancer mortality only among men in the highest intake quartile [Q1: reference; Q4: HR, 1.18 (95% CIs, 1.04–1.33)]. Higher pork consumption was also associated with an increased risk of total mortality in men [Q1: reference; Q4: HR, 1.11 (95% CIs, 1.01–1.22)]. In contrast, a moderate intake of processed meat was associated with a lower risk of total mortality [Q1: reference; Q2: HR, 0.91 (95% CIs: 0.85–0.98); Q3: 0.92 (95% CIs, 0.85–0.99)] and of cancer mortality [Q1: reference; Q2: HR, 0.88 (95% CI: 0.79–0.98); Q3: 0.89 (95% CIs, 0.79–0.99)] only among men, albeit that associations seen in Q3 were not significant after exclusion of deaths within 5 years. Higher intake of chicken was associated with a lower risk of cancer mortality in men [Q1: reference; Q4: HR, 0.85 (95% CIs: 0.75–0.96)]. This inverse association between chicken consumption and all-cause mortality was also seen after exclusion of deaths

**Table 2. Adjusted hazard ratios of mortality by meat consumption status (men).**

| | All-cause | | | Cancer | | | Cerebrovascular Disease | | | Heart Disease | | | Colorectal Cancer | | |
|---|---|---|---|---|---|---|---|---|---|---|---|---|---|---|---|
| | Cases | HR[1,2] | 95% CI | Cases | HR[1,2] | 95% CI | Cases | HR[1,2] | 95% CI | Cases | HR[1,2] | 95% CI | Cases | HR[1,2] | 95% CI |
| All meat | | | | | | | | | | | | | | | |
| Q1 | 1,788 | 1.00 | | 762 | 1.00 | | 173 | 1.00 | | 218 | 1.00 | | 79 | | |
| Q2 | 1,501 | 0.98 | (0.91–1.05) | 649 | 0.95 | (0.85–1.07) | 123 | 0.89 | (0.70–1.15) | 184 | 1.05 | (0.85–1.30) | 51 | 0.81 | (0.56–1.18) |
| Q3 | 1,420 | 0.99 | (0.91–1.07) | 640 | 1.00 | (0.88–1.12) | 130 | 1.03 | (0.78–1.35) | 156 | 0.99 | (0.78–1.26) | 38 | 0.63 | (0.40–0.98) |
| Q4 | 1,557 | **1.18** | **(1.06–1.31)** | 644 | 1.06 | (0.90–1.25) | 143 | 1.39 | (0.97–1.98) | 193 | **1.46** | **(1.08–1.99)** | 83 | 1.51 | (0.91–2.49) |
| *p for trend* | | **0.026** | | | 0.534 | | | 0.127 | | | 0.083 | | | 0.514 | |
| Red meat[3] | | | | | | | | | | | | | | | |
| Q1 | 1,819 | 1.00 | | 770 | 1.00 | | 174 | 1.00 | | 221 | 1.00 | | 75 | | |
| Q2 | 1,470 | 0.93 | (0.87–1.01) | 636 | 0.92 | (0.82–1.03) | 123 | 0.85 | (0.66–1.10) | 180 | 1.03 | (0.83–1.27) | 50 | 0.83 | (0.56–1.22) |
| Q3 | 1,430 | 0.98 | (0.90–1.06) | 647 | 1.01 | (0.89–1.14) | 130 | 1.02 | (0.78–1.35) | 153 | 1.00 | (0.78–1.28) | 46 | 0.83 | (0.54–1.27) |
| Q4 | 1,547 | **1.13** | **(1.02–1.26)** | 642 | 1.07 | (0.91–1.25) | 142 | 1.36 | (0.95–1.95) | 197 | **1.51** | **(1.11–2.06)** | 80 | 1.52 | (0.91–2.54) |
| *p for trend* | | 0.079 | | | 0.380 | | | 0.137 | | | **0.048** | | | 0.265 | |
| Beef[4] | | | | | | | | | | | | | | | |
| Q1 | 1,883 | 1.00 | | 747 | 1.00 | | 165 | 1.00 | | 264 | 1.00 | | 74 | | |
| Q2 | 1,515 | 1.03 | (0.59–1.79) | 666 | 1.08 | (0.96–1.20) | 146 | 1.22 | (0.96–1.55) | 178 | 0.85 | (0.70–1.04) | 59 | 1.04 | (0.72–1.49) |
| Q3 | 1,399 | 0.76 | (0.39–1.49) | 621 | 1.07 | (0.95–1.20) | 135 | 1.24 | (0.96–1.59) | 148 | 0.76 | (0.61–0.94) | 52 | 0.91 | (0.62–1.36) |
| Q4 | 1,469 | 1.12 | (0.50–2.48) | 661 | **1.18** | **(1.04–1.33)** | 123 | 1.12 | (0.84–1.50) | 161 | 0.83 | (0.66–1.06) | 66 | 1.19 | (0.79–1.78) |
| *p for trend* | | 0.562 | | | **0.019** | | | 0.312 | | | 0.051 | | | 0.563 | |
| Pork[5] | | | | | | | | | | | | | | | |
| Q1 | 1,744 | 1.00 | | 728 | 1.00 | | 190 | 1.00 | | 213 | 1.00 | | 67 | | |
| Q2 | 1,406 | 0.95 | (0.88–1.02) | 638 | 1.01 | (0.90–1.13) | 112 | **0.69** | **(0.54–0.89)** | 151 | 0.89 | (0.71–1.11) | 51 | 0.99 | (0.67–1.47) |
| Q3 | 1,475 | 0.98 | (0.91–1.06) | 647 | 1.02 | (0.90–1.15) | 122 | **0.74** | **(0.57–0.97)** | 185 | 1.13 | (0.90–1.42) | 54 | 1.06 | (0.70–1.60) |
| Q4 | 1,641 | **1.11** | **(1.01–1.22)** | 682 | 1.12 | (0.97–1.29) | 145 | 0.92 | (0.67–1.25) | 202 | 1.26 | (0.96–1.65) | 79 | 1.52 | (0.95–2.42) |
| *p for trend* | | 0.061 | | | 0.192 | | | 0.441 | | | 0.051 | | | 0.103 | |
| Processed meat[6] | | | | | | | | | | | | | | | |
| Q1 | 2,016 | 1.00 | | 850 | 1.00 | | 206 | 1.00 | | 247 | 1.00 | | 85 | | |
| Q2 | 1,471 | **0.91** | **(0.85–0.98)** | 628 | **0.88** | **(0.79–0.98)** | 128 | 0.80 | (0.63–1.02) | 183 | 1.02 | (0.83–1.25) | 55 | 0.86 | (0.60–1.24) |
| Q3 | 1,388 | **0.92** | **(0.85–0.99)** | 595 | **0.89** | **(0.79–0.99)** | 128 | 0.86 | (0.67–1.09) | 157 | 0.93 | (0.75–1.16) | 41 | 0.63 | (0.42–0.95) |
| Q4 | 1,391 | 0.98 | (0.91–1.07) | 622 | 1.00 | (0.88–1.13) | 107 | 0.85 | (0.64–1.13) | 164 | 1.05 | (0.83–1.32) | 70 | 1.04 | (0.71–1.55) |
| *p for trend* | | 0.488 | | | 0.759 | | | 0.251 | | | 0.935 | | | 0.751 | |
| Chicken[7] | | | | | | | | | | | | | | | |
| Q1 | 1,793 | 1.00 | | 769 | 1.00 | | 160 | 1.00 | | 230 | 1.00 | | 79 | | |
| Q2 | 1,504 | 0.97 | (0.90–1.04) | 678 | 0.98 | (0.88–1.09) | 129 | 0.91 | (0.71–1.16) | 183 | 0.96 | (0.79–1.18) | 60 | 0.93 | (0.65–1.32) |
| Q3 | 1,454 | 0.95 | (0.88–1.02) | 647 | 0.96 | (0.85–1.07) | 141 | 1.05 | (0.82–1.34) | 157 | 0.82 | (0.65–1.02) | 53 | 0.82 | (0.56–1.19) |
| Q4 | 1,515 | 0.94 | (0.87–1.02) | 601 | **0.85** | **(0.75–0.96)** | 139 | 1.01 | (0.77–1.31) | 181 | 0.92 | (0.73–1.15) | 59 | 0.85 | (0.58–1.25) |
| *p for trend* | | 0.128 | | | **0.010** | | | 0.742 | | | 0.245 | | | 0.329 | |

Abbreviations: HR, hazard ratio; 95% CI, 95% confidence intervals; Q, quartile.

[1] Cox proportional hazard models were used.

[2] Adjusted for age (years, continuous); public health center area; smoking status (never, former, current), alcohol intake (none, >0-<150 g/w, 150-<300 g/w, 300+g/w), BMI (<25, 25 - <27, 27-<30, 30+), quartile of metabolic equivalent task-hours/d, history of hypertension, history of diabetes, total energy intake, intakes of fruits, vegetables, fish, dairy products, egg, sodium and total fat (continuous).

[3] Additionally adjusted for intake of chicken.

[4] Additionally adjusted for intakes of pork, processed meat and chicken.

[5] Additionally adjusted for intakes of beef, processed meat and chicken.

[6] Additionally adjusted for intakes of beef, pork and chicken.

[7] Additionally adjusted for intake of red meat.

**Table 3. Adjusted hazard ratios of mortality by meat consumption status (women).**

| | All-cause | | | Cancer | | | Cerebrovascular Disease | | | Heart Disease | | | Colorectal Cancer | | |
|---|---|---|---|---|---|---|---|---|---|---|---|---|---|---|---|
| | Cases | HR[1,2] | 95% CI | Cases | HR[1,2] | 95% CI | Cases | HR[1,2] | 95% CI | Cases | HR[1,2] | 95% CI | Cases | HR[1,2] | 95% CI |
| All meat | | | | | | | | | | | | | | | |
| Q1 | 1,013 | 1.00 | | 404 | 1.00 | | 123 | 1.00 | | 128 | 1.00 | | 47 | | |
| Q2 | 850 | 0.98 | (0.89–1.08) | 354 | 1.01 | (0.87–1.17) | 79 | **0.70** | **(0.51–0.94)** | 106 | 0.99 | (0.75–1.30) | 43 | 1.08 | (0.69–1.67) |
| Q3 | 834 | 1.02 | (0.91–1.13) | 367 | 1.13 | (0.96–1.33) | 80 | **0.68** | **(0.50–0.95)** | 97 | 0.92 | (0.69–1.24) | 55 | 1.48 | (0.94–2.34) |
| Q4 | 923 | 1.11 | (0.97–1.26) | 354 | 1.18 | (0.96–1.45) | 89 | **0.66** | **(0.44–0.99)** | 127 | 1.09 | (0.77–1.56) | 49 | 1.45 | (0.81–2.60) |
| *p for trend* | | 0.164 | | | 0.070 | | | **0.029** | | | 0.829 | | | 0.104 | |
| Red meat[3] | | | | | | | | | | | | | | | |
| Q1 | 1,018 | 1.00 | | 405 | 1.00 | | 120 | 1.00 | | 130 | 1.00 | | 47 | | |
| Q2 | 834 | 0.96 | (0.87–1.06) | 356 | 1.01 | (0.87–1.18) | 90 | 0.84 | (0.63–1.13) | 100 | 0.91 | (0.69–1.21) | 44 | 1.11 | (0.71–1.73) |
| Q3 | 852 | 1.04 | (0.94–1.16) | 355 | 1.11 | (0.94–1.32) | 78 | **0.68** | **(0.49–0.96)** | 105 | 0.98 | (0.73–1.33) | 51 | 1.41 | (0.88–2.25) |
| Q4 | 916 | 1.08 | (0.95–1.24) | 363 | 1.23 | (0.99–1.51) | 83 | **0.62** | **(0.41–0.95)** | 123 | 1.01 | (0.70–1.45) | 52 | 1.52 | (0.84–2.74) |
| *p for trend* | | 0.174 | | | 0.051 | | | **0.013** | | | 0.921 | | | 0.111 | |
| Beef[4] | | | | | | | | | | | | | | | |
| Q1 | 1,136 | 1.00 | | 426 | 1.00 | | 128 | 1.00 | | 163 | 1.00 | | 57 | | |
| Q2 | 808 | 0.93 | (0.84–1.02) | 348 | 0.96 | (0.82–1.11) | 91 | 0.96 | (0.71–1.28) | 88 | 0.76 | (0.58–1.01) | 46 | 1.07 | (0.71–1.63) |
| Q3 | 812 | 0.96 | (0.87–1.06) | 356 | 1.04 | (0.89–1.21) | 81 | 0.87 | (0.64–1.18) | 96 | 0.89 | (0.68–1.18) | 52 | 1.13 | (0.75–1.72) |
| Q4 | 864 | 1.00 | (0.91–1.11) | 349 | 1.05 | (0.89–1.24) | 71 | 0.71 | (0.51–1.00) | 111 | 0.95 | (0.72–1.25) | 39 | 0.88 | (0.55–1.41) |
| *p for trend* | | 0.914 | | | 0.415 | | | 0.050 | | | 0.777 | | | 0.748 | |
| Pork[5] | | | | | | | | | | | | | | | |
| Q1 | 979 | 1.00 | | 415 | 1.00 | | 105 | 1.00 | | 118 | 1.00 | | 46 | | |
| Q2 | 809 | 0.93 | (0.84–1.03) | 339 | 0.90 | (0.77–1.05) | 79 | 0.79 | (0.57–1.09) | 106 | 1.03 | (0.78–1.37) | 46 | 1.20 | (0.77–1.87) |
| Q3 | 866 | 0.99 | (0.89–1.10) | 348 | 0.95 | (0.81–1.12) | 91 | 0.95 | (0.69–1.31) | 95 | 0.85 | (0.63–1.15) | 46 | 1.13 | (0.71–1.82) |
| Q4 | 966 | 0.97 | (0.86–1.10) | 377 | 1.03 | (0.85–1.25) | 96 | 0.80 | (0.54–1.17) | 139 | 0.96 | (0.69–1.34) | 56 | 1.37 | (0.80–2.33) |
| *p for trend* | | 0.887 | | | 0.766 | | | 0.436 | | | 0.546 | | | 0.326 | |
| Processed meat[6] | | | | | | | | | | | | | | | |
| Q1 | 1,145 | 1.00 | | 451 | 1.00 | | 115 | 1.00 | | 149 | 1.00 | | 62 | | |
| Q2 | 825 | 0.93 | (0.84–1.02) | 341 | 0.90 | (0.77–1.04) | 91 | 1.13 | (0.84–1.52) | 96 | 0.92 | (0.70–1.21) | 35 | 0.68 | (0.44–1.07) |
| Q3 | 814 | 0.98 | (0.89–1.08) | 333 | 0.95 | (0.81–1.11) | 89 | 1.18 | (0.87–1.61) | 106 | 1.08 | (0.82–1.42) | 39 | 0.83 | (0.53–1.28) |
| Q4 | 836 | 1.05 | (0.95–1.17) | 354 | 1.10 | (0.93–1.30) | 76 | 1.01 | (0.71–1.42) | 107 | 1.11 | (0.83–1.48) | 58 | 1.26 | (0.82–1.94) |
| *p for trend* | | 0.327 | | | 0.266 | | | 0.793 | | | 0.355 | | | 0.260 | |
| Chicken[7] | | | | | | | | | | | | | | | |
| Q1 | 1,008 | 1.00 | | 409 | 1.00 | | 97 | 1.00 | | 132 | 1.00 | | 50 | | |
| Q2 | 863 | 1.00 | (0.91–1.10) | 370 | 0.99 | (0.86–1.15) | 90 | 1.05 | (0.77–1.42) | 107 | 1.02 | (0.78–1.33) | 51 | 1.12 | (0.74–1.69) |
| Q3 | 834 | 0.96 | (0.87–1.06) | 359 | 0.98 | (0.84–1.15) | 82 | 0.95 | (0.69–1.31) | 91 | 0.89 | (0.67–1.18) | 41 | 0.92 | (0.59–1.43) |
| Q4 | 915 | 1.00 | (0.90–1.10) | 341 | 0.94 | (0.79–1.10) | 102 | 1.04 | (0.75–1.44) | 128 | 1.08 | (0.81–1.42) | 52 | 1.09 | (0.70–1.70) |
| *p for trend* | | 0.789 | | | 0.439 | | | 0.964 | | | 0.819 | | | 0.913 | |

Abbreviations: HR, hazard ratio; 95% CI, 95% confidence intervals; Q, quartile.

[1] Cox proportional hazard models were used.

[2] Adjusted for age (years, continuous); public health center area; smoking status (never, former, current), alcohol intake (none, >0-<150 g/w, 150-<300 g/w, 300+g/w), BMI (<25, 25 - <27, 27-<30, 30+), quartile of metabolic equivalent task-hours/d, history of hypertension, total energy intake, history of diabetes, intakes of fruits, vegetables, fish, dairy products, egg, sodium and total fat (continuous).

[3] Additionally adjusted for intake of chicken.

[4] Additionally adjusted for intakes of pork, processed meat and chicken.

[5] Additionally adjusted for intakes of beef, processed meat and chicken.

[6] Additionally adjusted for intakes of beef, pork and chicken.

[7] Additionally adjusted for intake of red meat.

occurring within 5 years [Q1: reference, Q4: HR, 0.83 (95% CIs: 0.72–0.96)]. No significant associations were observed between all types of meat and colorectal cancer mortality. In our sub-analysis, no significant association was seen between meat intake and mortality due to ischemic heart disease or intracerebral hemorrhage (S3 Table). Across other types of cause-specific mortality, the same associations remained after exclusion of deaths that occurred within 5 years after the 5-year follow-up survey in both men and women.

Table 4 shows the results of meat intake and all-cause and cause-specific mortality according to age groups at baseline (45–54 years, 55–64 years, and 65 years and older) in men and women, respectively. In men, a significant increase in all-cause mortality risk was observed in those older than 65 years [Q1: reference; Q4: HR, 1.21 (95% CIs: 1.03–1.41)]. Red meat intake was also associated with a higher risk of all-cause mortality in those aged 55–64 years [Q1: reference; Q4: HR, 1.20 (95% CIs: 1.02–1.42)]. The risk difference was significant by age group (p-interaction with age: total meat, 0.004; red meat, 0.018). In women, a significant increase in all-cause mortality risk was seen in those aged 45–54 years who consumed more total meat [Q1: reference; Q2: HR, 1.32 (95% CIs: 1.01–1.72); Q3: HR, 1.37 (95% CIs: 1.03–1.82)] and more red meat [Q1: reference; Q3: HR, 1.34 (95% CIs: 1.01–1.79)].

## Discussion

Our study is one of the few conducted in Asia to assess the association between meat intake and mortality due to all-cause and leading causes using data from a large-scale prospective study. Compared with subjects in the lowest quartile of total meat intake, men in the highest quartile had 18% higher risk of dying from all causes, although the associations were not dose-responsive. Our results agree with a 2014 meta-analysis of nine prospective cohort studies in the US, Europe and China, although total red meat intake in that review increased the risk of all-cause mortality by 29% in the highest intake category [10] compared with 18% in our study. This difference in the magnitude of risk increase is because the analyses in both studies were performed using relative consumption categories rather than absolute consumption amounts. The difference may also be due to the fact that the absolute amount of intake of meat differs between the Japanese and Western populations: mean daily intake of total meat in the US population amounted to 127.9 grams per day between 2003 and 2004 [25] versus 77 grams per day in Japan as of 2003 [26]. Mean intake amount in the US is almost twice that in Japan.

In our study population, the crude mean intake (adjusted for the difference between the FFQ and dietary records [20, 21]) of total meat was 71.9 grams per day.

We also noted generational differences in the association patterns between meat intake and all-cause mortality: notably, women aged 45–54 years had significantly higher risk of total mortality. However, these results could be a chance finding resulting from the multiple statistical tests conducted, given that the highest intake quartile showed no significant associations. In men, in contrast, elevated risk of mortality in the highest intake quartile was seen only in the older groups, and not in the younger age group.

Of note, our study found that intake of total meat was associated with a decreased risk of cerebrovascular disease mortality in women. Meat is a major source of animal protein, and a modest amount of protein intake has been reported to suppress blood pressure and thereby prevent stroke [27]. For instance, previous population-based studies—both derived from Japanese populations—showed that dietary animal protein intake was associated with lower blood pressure levels [28] and also a reduced risk of intraparenchymal hemorrhage [29]. Nevertheless, our results did not show any association between meat intake and intracerebral hemorrhage, a finding which warrants further investigation.

**Table 4. Adjusted hazard ratios of all-cause mortality according to quartile of meat consumption by age group.**

| | 45–54 years | | | 55–64 years | | | 65–79 years | | |
|---|---|---|---|---|---|---|---|---|---|
| | Cases | HR[1,2] | 95% CI | Cases | HR[1,2] | 95% CI | Cases | HR[1,2] | 95% CI |
| **Men** | | | | | | | | | |
| All meat | | | | | | | | | |
| Median intake (g/d)[3] | | 50.4 | | | 45.6 | | | 45.4 | |
| Q1 | 270 | 1.00 | | 766 | 1.00 | | 752 | 1.00 | |
| Q2 | 250 | 0.85 | (0.71–1.03) | 646 | 0.99 | (0.88–1.10) | 605 | 1.01 | (0.90–1.13) |
| Q3 | 260 | 0.96 | (0.78–1.17) | 555 | 0.96 | (0.84–1.09) | 605 | 1.02 | (0.90–1.16) |
| Q4 | 249 | 1.10 | (0.84–1.44) | 626 | 1.17 | (0.99–1.38) | 682 | **1.21** | **(1.03–1.41)** |
| *p-interaction with age*: | | **0.004** | | | | | | | |
| Red meat[4] | | | | | | | | | |
| Median intake (g/d)[3] | | 42.3 | | | 37.9 | | | 37.7 | |
| Q1 | 271 | 1.00 | | 773 | 1.00 | | 775 | 1.00 | |
| Q2 | 253 | 0.87 | (0.73–1.05) | 626 | 0.96 | (0.86–1.08) | 591 | 0.92 | (0.82–1.04) |
| Q3 | 259 | 0.99 | (0.80–1.21) | 564 | 0.99 | (0.87–1.13) | 607 | 0.95 | (0.84–1.08) |
| Q4 | 246 | 1.12 | (0.85–1.47) | 630 | **1.20** | **(1.02–1.42)** | 671 | 1.08 | (0.92–1.27) |
| *p-interaction with age*: | | **0.018** | | | | | | | |
| Chicken[5] | | | | | | | | | |
| Median intake (g/d)[3] | | 5.8 | | | 5.1 | | | 5.0 | |
| Q1 | 260 | 1.00 | | 751 | 1.00 | | 782 | 1.00 | |
| Q2 | 294 | 1.04 | (0.88–1.24) | 653 | 0.91 | (0.82–1.02) | 557 | 0.99 | (0.89–1.11) |
| Q3 | 236 | 0.83 | (0.69–1.01) | 602 | **0.88** | **(0.78–0.99)** | 616 | 1.09 | (0.97–1.22) |
| Q4 | 239 | 0.90 | (0.74–1.11) | 587 | 0.92 | (0.82–1.04) | 689 | 0.99 | (0.88–1.11) |
| *p-interaction with age*: | | **0.002** | | | | | | | |
| **Women** | | | | | | | | | |
| All meat | | | | | | | | | |
| Median intake (g/d)[3] | | 50.9 | | | 45.2 | | | 44.9 | |
| Q1 | 101 | 1.00 | | 365 | 1.00 | | 547 | 1.00 | |
| Q2 | 144 | **1.32** | **(1.01–1.72)** | 311 | 0.95 | (0.81–1.12) | 395 | 0.95 | (0.82–1.08) |
| Q3 | 154 | **1.37** | **(1.03–1.82)** | 299 | 1.05 | (0.88–1.25) | 381 | 0.92 | (0.79–1.06) |
| Q4 | 155 | 1.41 | (0.99–2.01) | 319 | 1.19 | (0.95–1.49) | 449 | 0.99 | (0.83–1.19) |
| *p-interaction with age*: | | 0.093 | | | | | | | |
| Red meat[4] | | | | | | | | | |
| Median intake (g/d)[3] | | 42.0 | | | 37.3 | | | 37.0 | |
| Q1 | 106 | 1.00 | | 373 | 1.00 | | 539 | 1.00 | |
| Q2 | 139 | 1.20 | (0.91–1.56) | 311 | 0.90 | (0.77–1.06) | 384 | 0.96 | (0.83–1.10) |
| Q3 | 155 | **1.34** | **(1.01–1.79)** | 290 | 1.00 | (0.83–1.20) | 407 | 1.00 | (0.86–1.17) |
| Q4 | 154 | 1.37 | (0.95–1.96) | 320 | 1.15 | (0.92–1.45) | 442 | 0.97 | (0.81–1.17) |
| *p-interaction with age*: | | 0.196 | | | | | | | |
| Chicken[5] | | | | | | | | | |
| Median intake (g/d)[3] | | 6.7 | | | 5.9 | | | 5.6 | |
| Q1 | 116 | 1.00 | | 338 | 1.00 | | 554 | 1.00 | |
| Q2 | 151 | 1.07 | (0.83–1.38) | 337 | 1.05 | (0.90–1.23) | 375 | 0.93 | (0.81–1.07) |
| Q3 | 135 | 0.88 | (0.67–1.15) | 322 | 1.00 | (0.85–1.18) | 377 | 0.97 | (0.84–1.11) |
| Q4 | 152 | 0.97 | (0.73–1.27) | 297 | 0.95 | (0.80–1.14) | 466 | 1.03 | (0.89–1.18) |

*(Continued)*

**Table 4.** (Continued)

| | 45–54 years | | | 55–64 years | | | 65–79 years | | |
|---|---|---|---|---|---|---|---|---|---|
| | Cases | HR[1,2] | 95% CI | Cases | HR[1,2] | 95% CI | Cases | HR[1,2] | 95% CI |
| *p-interaction with age*: | | 0.868 | | | | | | | |

Abbreviations: HR, hazard ratio; 95% CI, 95% confidence intervals; Q, quartile.

[1] Cox proportional hazard models were used.

[2] Adjusted for age (years, continuous); public health center area; smoking status (never, former, current), alcohol intake (none, >0-<150 g/w, 150-<300 g/w, 300+g/w), BMI (<25, 25-<27, 27-<30, 30+), quartile of metabolic equivalent task-hours/d, history of hypertension, history of diabetes, total energy intake, intakes of fruits, vegetables, fish, dairy products, egg, sodium and total fat (continuous).

[3] All median intakes of meat are energy-adjusted.

[4] Additionally adjusted for intake of chicken.

[5] Additionally adjusted for intake of red meat.

On the contrary, our study reported an elevated risk of heart disease mortality in men but not in women if red meat is taken to the level of the highest intake quartile. One possible mechanism of this association between red meat and heart disease is that meat is a major source of saturated fatty acid, which is reported to increase the risk of myocardial infarction [30]. Several cohort studies have also reported an association between red meat consumption and cardiovascular disease mortality [5, 31], although we did not observe any association between red meat intake and ischemic heart disease mortality in the current study.

Our study also found an elevated risk of cancer mortality in men with the highest intake of beef. Although not specifically referring to beef, the World Cancer Research Fund reported that red meat increases the risk of colorectal cancer [32], and showed suggestive evidence that red meat increases the risk of nasopharynx, lung and pancreatic cancer [32]. The International Agency for Research on Cancer (IARC) also reported that red meat intake is associated with the risk of prostate cancer [33]. Potential pathways can be explained as follows: in the process of cooking red meat, the nitrites convert to *N*-nitroso compounds, which are known carcinogens that function as a series of initiators and promoters in cancer [34–36]. Heme iron from red meat can catalyze lipid peroxidation, and cause DNA damage in tissues [37, 38]. Other carcinogens, including heterocyclic aromatic amines and polycyclic aromatic hydrocarbons, which are produced by cooking of meat at high temperatures (e.g. pan-frying, grilling or barbecuing), also contribute to carcinogenesis of different sites [33, 39–42]. Despite mounting evidence linking red meat intake and colorectal cancer incidence globally [32] and in Japan [23], no association was seen between meat and colorectal cancer mortality in our study; this may have been due to the fact that the survival rate of colorectal cancer is generally high in Japan, for example with a 5-year relative survival rate of 71.1% for cases diagnosed between 2006–2008 [43, 44].

Contrary to previous studies [5, 45], our analysis showed that processed meat consumption was not associated with elevated risk of mortality due to all-cause, cancer, heart disease and cerebrovascular disease. This null association might be explained by the difference in the intake amount of processed meat in Japan: generally, consumption is lower in Japan than in Western countries. For example, average consumption in the US was estimated to be 23.2 grams per day between 2003 and 2004 [25] versus 12 grams per day in Japan as of 2003 [26]. In our study population, the crude mean intake (adjusted for the difference between the FFQ and dietary records [46]) of processed meat was 7.7 grams per day. Such difference in the absolute amount intake of processed meat may have attenuated the adverse effects.

While a moderate intake of processed meat and pork showed a lower risk of mortality in men, this may have been a chance finding as we did not see any significant trend for different quartile intakes of the corresponding meat type. In contrast, our results showed that higher intake of white meat (i.e. chicken) is associated with a modest decrease in the risk of cancer mortality while red meat intake was held constant. A US study reported that poultry intake was inversely associated with esophagus squamous cell carcinoma, liver cancer and lung cancer, assuming no change in the amount intake of red meat [47]. Given that both studies adjusted for red meat intake, the inverse associations are not due to the effect of substituting red meat with poultry. The mechanism whereby chicken intake alone reduces cancer risk is unclear, and further research is required in this area.

The major strength of this study is the fact that the subjects were recruited from a large sample of the Japanese population. The high response rate and low loss to follow-up may have reduced selection bias. With an average of 14.0 years of follow-up, we consider that sufficient numbers of deaths due to all-cause, overall cancer, heart disease and cerebrovascular disease were captured. Second, we used a 5-year follow-up survey containing the FFQ collecting the necessary information about daily diet with reasonable degree of validity. On the other hand, some limitations warrant mention. Given the information on meat intake was collected only at 5-year follow-up, any change in meat intake during follow-up after the five-year follow-up survey may have produced misclassification.

## Conclusion

In this prospective, large-scale cohort study, heavy intake of total and red meat was associated with an increased risk of total mortality in men. In contrast, modest intake of total meat was associated with a lower risk of cerebrocascular disease mortality in women.

## Supporting information

**S1 Fig. Participant flow chart.**
(TIF)

**S1 Table. Adjusted hazard ratios of mortality by meat consumption status after excluding deaths within 5 years (men).**
(DOCX)

**S2 Table. Adjusted hazard ratios of mortality by meat consumption status after excluding deaths within 5 years (women).**
(DOCX)

**S3 Table. Adjusted hazard ratios of mortality from ischemic heart disease and intracerebral hemorrhage by meat consumption status.**
(DOCX)

## Acknowledgments

ES and XT analyzed the data, drafted the manuscript, reviewed and edited the manuscript, and contributed to discussion; MI and ST conducted, designed, and supervised the study, reviewed and edited the manuscript, and contributed to discussion; SKA, NS, JI, RT, HI, TS, TY and MIw reviewed and edited the manuscript, and contributed to discussion. All authors read and approved the final manuscript. Members of the Japan Public Health Center-based Prospective Study (JPHC Study; principal investigator, S. Tsugane; e-mail, stsugane@ncc.go.jp) Group are: S. Tsugane, N. Sawada, M. Iwasaki, M. Inoue, T. Yamaji, R. Katagiri, T. Imatoh, H. Ihira, S. K.

Abe and S. Tanaka, National Cancer Center, Tokyo; T. Sugie, Iwate Prefectural Ninohe Public Health Center, Iwate; T. Minamizono, Akita Prefectural Yokote Public Health Center, Akita; Y. Shirai, Nagano Prefectural Saku Public Health Center, Nagano; H. Sakiyama, Okinawa Prefectural Chubu Public Health Center, Okinawa; T. Yoshimi, Ibaraki Prefectural Chuo Public Health Center, Ibaraki; H. Sonoda, Niigata Prefectural Nagaoka Public Health Center, Niigata; T. Tagami, Kochi Prefectural Chuo-higashi Public Health Center, Kochi; T. Ando, Nagasaki Prefectural Kamigoto Public Health Center, Nagasaki; Y. Miyasato, Okinawa Prefectural Miyako Public Health Center, Okinawa; Y. Kokubo, National Cerebral and Cardiovascular Center, Osaka; K. Yamagishi, University of Tsukuba, Ibaraki; T. Mizoue, National Center for Global Health and Medicine, Tokyo; K. Nakamura, Niigata University, Niigata; and R. Takachi, Nara Women's University, Nara; J. Ishihara, Azabu University, Kanagawa; H. Iso and T. Kitamura, Osaka University, Osaka; I. Saito, Oita University, Oita; N. Yasuda, Kochi University, Kochi; M. Mimura, Keio University, Tokyo; K. Sakata, Iwate Medical University, Iwate; M. Noda, Saitama Medical University; A. Goto, Yokohama City University, Kanagawa; H. Yatsuya, Fujita Health University, Aichi.

## Author Contributions

**Conceptualization:** Manami Inoue, Shoichiro Tsugane.

**Formal analysis:** Eiko Saito, Xiaohe Tang.

**Methodology:** Eiko Saito, Xiaohe Tang.

**Supervision:** Manami Inoue, Shoichiro Tsugane.

**Writing – original draft:** Eiko Saito, Xiaohe Tang.

**Writing – review & editing:** Eiko Saito, Sarah Krull Abe, Norie Sawada, Junko Ishihara, Ribeka Takachi, Hiroyasu Iso, Taichi Shimazu, Taiki Yamaji, Motoki Iwasaki, Manami Inoue, Shoichiro Tsugane.

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
