## [Decision Letter · Decision Letter 0]

25 Sep 2020

PONE-D-20-16630

Association between meat intake and mortality due to all-cause and major causes of death in a Japanese population

PLOS ONE

Dear Dr. Inoue,

Thank you for submitting your manuscript to PLOS ONE. After careful consideration, we feel that it has merit but does not fully meet PLOS ONE’s publication criteria as it currently stands. Therefore, we invite you to submit a revised version of the manuscript that addresses the points raised during the review process.

We look forward to receiving your revised manuscript.

Kind regards,

Marly A. Cardoso, Ph.D.

Academic Editor

PLOS ONE

Journal Requirements:

2.We note that you have indicated that data from this study are available upon request. PLOS only allows data to be available upon request if there are legal or ethical restrictions on sharing data publicly. For information on unacceptable data access restrictions, please see http://journals.plos.org/plosone/s/data-availability#loc-unacceptable-data-access-restrictions.

3.One of the noted authors is a group or consortium [JPHC Study Group]. In addition to naming the author group, please list the individual authors and affiliations within this group in the acknowledgments section of your manuscript. Please also indicate clearly a lead author for this group along with a contact email address.

Additional Editor Comments (if provided):

The study is well-organized with interesting results. However, the manuscript needs revision following all reviewer´s suggestions. Particular attention should be given to the comparison on absolute meat intakes in Western and Asian populations, providing mean or median meat intake ranges in this study and also across populations.

Reviewers' comments:

Reviewer's Responses to Questions

**Comments to the Author**

1. Is the manuscript technically sound, and do the data support the conclusions?

Reviewer #1: Partly

2. Has the statistical analysis been performed appropriately and rigorously? 

Reviewer #1: Yes

3. Have the authors made all data underlying the findings in their manuscript fully available?

Reviewer #1: No

4. Is the manuscript presented in an intelligible fashion and written in standard English?

Reviewer #1: Yes

5. Review Comments to the Author

Reviewer #1: The authors have presented findings from a large cohort followed for a reasonable length of time when considering associations between dietary intake and mortality. This study is of interest to the scientific community in light of the landmark publication from the EAT Lancet commission and "The Reference Diet" suggesting a lower meat intake for many Western populations for both health benefits and a sustainable food supply for the future whilst at the same time noting that some populations will need a higher intake of meat.

In the introduction the authors comment on the novelty of this finding as meat intake is studied in a predominantly Asian population with a markedly lower meat intake than a Western population such as the United States. Given the difference in intake in these populations I would have liked to see the authors make a greater note of comparison on meat intake in Western and Asian populations for their significant findings and also when comparing to other studies. Whilst this was noted for one finding, for the remainder it was lacking.

The first paragraph of the discussion section presents overal findings for an association with higher meat intake and increased risk for men AND women. The percentage figures presented in the discussion were not presented in the results or available on any of the submitted tables. The authors also indicate that there was an overall increased risk for women as well as men and this finding was not commented on in the results or evident from the data.

Other edits and comments outlined below:

Abstract

Pg 3, Line 49: re-word for improved clarity – suggest adding “higher intake”

Methods

Pg 7, Line 1: add reference for the validation study of the FFQ

Pg 7, line 4, correct spelling x 2 should read “fried” not “fired”

Pg 7, line 14/15 add in a descriptor of “fair-moderate” to describe the validity and reproducibility of the meat intake from the FFQ compared to the food record

Pg 8, line 14 provide a definition, description or reference for metabolic equivalent task-hours

Pg 10, line 4 remove the word “both”

Discussion

General comments: authors should comment on findings in comparison to non-Asian populations and with reference to absolute meat intakes. When comparing to other studies a comment on mean or median meat intake would be helpful as you have discussed a marked difference in Asian and Western populations with regards to meat intake in introduction

Pg 12, line 5 & 6: 17% and 15% does not correspond with any data presented in tables – is this another analysis that was not reported?? If yes include in results. From Table 1 Men had 18% increased risk of dying from all-causes and in women this was not significant.

Pg 12, line 14 – relate this comment on explaining differences in age groups to average intake in Western diet – 5g/day does not seem “considerably” higher in comparison to difference with Western intake. Also of note is that the highest quartiles was not significantly different from lowest quartile – only the middle quartiles. Perhaps chance finding? Comment on number of statistical tests conducted and chance findings

Pg 12, line 20 – Elevated risk in men was only significant in older age group not younger as seems to be indicated from this paragraph

Pg 13, line 2, 3 it is important to add in here that these studies you are referring to are also in Japanese Populations

Pg 13, line 8,9 add in that this finding was in men only not women

Pg 13, line 18 please use the most up to date references from the WCRF report on red meat and cancer instead of reference 30. The Third Expert Report on Diet, Nutrition, Physical Activity and Cancer was published in 2018

Pg 14, line 8,9 again the authors need to consider the difference in absolute intake of meat in their study population and Western populations as this may be a reason that no association was found.

Page 14, line 21, 22 – have the authors considered chance findings for other statistically significant results?

6. PLOS authors have the option to publish the peer review history of their article (what does this mean?). If published, this will include your full peer review and any attached files.

Reviewer #1: **Yes: **Siobhan Hickling

---

## [Author Response · Author response to Decision Letter 0]

8 Nov 2020

RESPONSE TO REVIEWS

Editor Comments:

Response: We thank the editor for these comments. We have revised the styles and file naming according to the instructions provided online. 

2. We note that you have indicated that data from this study are available upon request. PLOS only allows data to be available upon request if there are legal or ethical restrictions on sharing data publicly. For information on unacceptable data access restrictions, please see http://journals.plos.org/plosone/s/data-availability#loc-unacceptable-data-access-restrictions. In your revised cover letter, please address the following prompts:

Response: We thank the editor for these detailed instructions. Accordingly, we have added a data availability statement in the attached cover letter.

3. One of the noted authors is a group or consortium [JPHC Study Group]. In addition to naming the author group, please list the individual authors and affiliations within this group in the acknowledgments section of your manuscript. Please also indicate clearly a lead author for this group along with a contact email address.

Response: We thank the editor for these instructions. We have listed the individual authors and their affiliations within this group in the revised Acknowledgements section, along with information on the principal investigator.

4. Additional Editor Comments (if provided):

The study is well-organized with interesting results. However, the manuscript needs revision following all reviewer´s suggestions. Particular attention should be given to the comparison on absolute meat intakes in Western and Asian populations, providing mean or median meat intake ranges in this study and also across populations.

Response: We thank the editor for these comments. We have provided mean meat intake ranges in this study and comparison against Western populations throughout the revised manuscript. Specific revisions are described in the point-by-point answers below. 

Reviewer #1 comments:

1. In the introduction the authors comment on the novelty of this finding as meat intake is studied in a predominantly Asian population with a markedly lower meat intake than a Western population such as the United States. Given the difference in intake in these populations I would have liked to see the authors make a greater note of comparison on meat intake in Western and Asian populations for their significant findings and also when comparing to other studies. Whilst this was noted for one finding, for the remainder it was lacking.

Response: We thank the editor for these comments. We have provided median meat intake ranges in this study and comparison against the Western populations as below:

“The difference may also be due to the fact that the absolute amount of intake of meat differs between the Japanese and Western populations: mean daily intake of total meat in the US population amounted to 127.9 grams per day between 2003 and 2004 (25) versus 77 grams per day in Japan as of 2003 (26). Mean intake amount in the US is almost twice that in Japan. In our study population, the crude mean intake (adjusted for the difference between the FFQ and dietary records) of total meat was 71.9 grams per day.”(Line 279-285, page 20)

Also, we have added a discussion on processed meat intake as below:

“For example, average consumption in the US was estimated to be 23.2 grams per day between 2003 and 2004 versus 12 grams per day in Japan as of 2003. In our study population, the crude mean intake (adjusted for the difference between the FFQ and dietary records) of processed meat was 7.7 grams per day.” (Line 336-340, page 23)

2. The first paragraph of the discussion section presents overall findings for an association with higher meat intake and increased risk for men AND women. The percentage figures presented in the discussion were not presented in the results or available on any of the submitted tables. The authors also indicate that there was an overall increased risk for women as well as men and this finding was not commented on in the results or evident from the data.　 

Response: We thank the reviewer for pointing out this discrepancy, which occurred during the course of re-analyses. We have corrected the corresponding sentence, and the revised manuscript now reads as below:

“Compared with subjects in the lowest quartile of total meat intake, men in the highest quartile had 18% higher risk of dying from all causes, although the associations were not dose-responsive.” (Line 272-274, page 20)

3. Abstract Pg 3, Line 49: re-word for improved clarity – suggest adding “higher intake”

Response: We have added “higher intake” in the revised manuscript. (line 34, page 2)

4. Methods Pg 7, Line 1: add reference for the validation study of the FFQ 

Response: Accordingly, we have added the references for the validation studies of the FFQ. (Line 120, page 6)

5. Pg 7, line 4, correct spelling should read “fried” not “fired”

Response: Accordingly, we have corrected this spelling error.

6. Pg 7, line 14/15 add in a descriptor of “fair-moderate” to describe the validity and reproducibility of the meat intake from the FFQ compared to the food record. 

Response: We have added “The validity of the FFQ for the assessment of meat intake has been previously reported as fair to moderate” in the revised manuscript. (Line 131-132, page 6)

7. Pg 8, line 14 provide a definition, description or reference for metabolic equivalent task-hours 

Response: We have added a reference for metabolic equivalent task-hours in the revised manuscript. (Line 154, page 7)

8. Pg 10, line 4 remove the word “both”

Response: We have removed the word “both” in the revised manuscript.

9. Discussion, General comments: authors should comment on findings in comparison to non-Asian populations and with reference to absolute meat intakes. When comparing to other studies a comment on mean or median meat intake would be helpful as you have discussed a marked difference in Asian and Western populations with regards to meat intake in introduction 

Response: We thank the reviewer for the comment. We have provided mean meat intake ranges in this study and comparison against the Western populations as shown in our response to reviewer comment #1.

10. Pg 12, line 5 & 6: 17% and 15% does not correspond with any data presented in tables – is this another analysis that was not reported?? If yes include in results.

Response: Again, we thank the reviewer for pointing out this discrepancy, which occurred during the course of re-analyses. We have corrected the corresponding sentence, and the revised manuscript now reads as below:

“Compared with subjects in the lowest quartile of total meat intake, men in the highest quartile had 18% higher risk of dying from all causes, although the associations were not dose-responsive.” (Line 272-274, page 20)

11. From Table 1 Men had 18% increased risk of dying from all-causes and in women this was not significant.

Response: We thank the reviewer for pointing out this discrepancy, which occurred during the course of re-analyses. We have corrected the corresponding description as indicated in our response to comment #10 (above).

12. Pg 12, line 14 – relate this comment on explaining differences in age groups to average intake in Western diet – 5g/day does not seem “considerably” higher in comparison to difference with Western intake. Also of note is that the highest quartiles was not significantly different from lowest quartile – only the middle quartiles. Perhaps chance finding? Comment on number of statistical tests conducted and chance findings.

Response: We thank the reviewer for this valuable input. We agree that the results for women could be due to a chance finding, since the highest intake quartile did not show any significant association, as the reviewer indicated. The revised manuscript thus reads as below:

“We also noted generational differences in the association patterns between meat intake and all-cause mortality: notably, women aged 45-54 years had significantly higher risk of total mortality. However, these results could be a chance finding resulting from the multiple statistical tests conducted, given that the highest intake quartile showed no significant associations. (Line 287-291, page 20-21)

13. Pg 12, line 20 – Elevated risk in men was only significant in older age group not younger as seems to be indicated from this paragraph 

Response: Yes, we have revised the corresponding sentence to improve the clarity of the manuscript as below: 

“In men, in contrast, elevated risk of mortality in the highest intake quartile was seen only in the older groups, and not in the younger age group.” (Line 291-292, page 21)

14. Pg 13, line 2, 3 it is important to add in here that these studies you are referring to are also in Japanese Populations

Response: We thank the reviewer for this valuable comment. We have added corresponding text in the revised manuscript:

“For instance, previous population-based studies – both derived from Japanese populations – showed that dietary animal protein intake was associated with lower blood pressure levels and also a reduced risk of intraparenchymal hemorrhage.” (Line 297-300, page 21)

15. Pg 13, line 8,9 add in that this finding was in men only not women 

Response: Accordingly, we have revised this text as below:

“On the contrary, our study reported an elevated risk of heart disease mortality in men but not in women if red meat is taken to the level of the highest intake quartile.” (Line 304-305, page 21)

16. Pg 13, line 18 please use the most up to date references from the WCRF report on red meat and cancer instead of reference The Third Expert Report on Diet, Nutrition, Physical Activity and Cancer was published in 2018.

Response: The Third Expert Report on Diet, Nutrition, Physical Activity and Cancer published in 2018 is the most recent report published to date by the WCRF Continuous Update Project (https://www.wcrf.org/int/continuous-update-project). Given the comprehensiveness of the report covering all major cancer sites and risk factors, and because of the robustness of the methodology, we thought that the 2018 report is appropriate to reference in our manuscript. 

17. Pg 14, line 8,9 again the authors need to consider the difference in absolute intake of meat in their study population and Western populations as this may be a reason that no association was found.

Response: Following reviewer’s recommendation, we have added a description of the mean intake of processed meat in the revised manuscript as below:

“In our study population, the crude mean intake (adjusted for the difference between the FFQ and dietary records) of processed meat was 7.7 grams per day.” (Line 338-340, page 23)

18. Page 14, line 21, 22 – have the authors considered chance findings for other statistically significant results?

Response: We have carefully checked the possibility of chance findings for all meat types and outcomes, and cross-checked the results with p-values for trends, along with our sensitivity analyses which excluded deaths within 5 years. As for chance finding in the age-group analyses, we have added the possibility of a chance finding as suggested by the reviewer’s comment #12.

---

## [Editor Report · Decision Letter 1]

2 Dec 2020

Association between meat intake and mortality due to all-cause and major causes of death in a Japanese population

PONE-D-20-16630R1

Dear Dr. Inoue,

We’re pleased to inform you that your manuscript has been judged scientifically suitable for publication and will be formally accepted for publication once it meets all outstanding technical requirements.

Kind regards,

Marly A. Cardoso, Ph.D.

Academic Editor

PLOS ONE
---

## [Editor Report · Acceptance letter]

4 Dec 2020

PONE-D-20-16630R1 

Association between meat intake and mortality due to all-cause and major causes of death in a Japanese population. 

Dear Dr. Inoue:

I'm pleased to inform you that your manuscript has been deemed suitable for publication in PLOS ONE. Congratulations! Your manuscript is now with our production department. 

Kind regards, 

on behalf of

Dr. Marly A. Cardoso 

Academic Editor

PLOS ONE